# An intronic copy number variation in *Syntaxin 17* determines speed of greying and melanoma incidence in Grey horses

Carl-Johan Rubin [1,2], McKaela Hodge [3,12], Rakan Naboulsi [4,11,12], Madeleine Beckman [5], Rebecca R. Bellone [6,7], Angelica Kallenberg [6], Stephanie J'Usrey [6], Hajime Ohmura [8], Kazuhiro Seki [9], Risako Furukawa [10], Aoi Ohnuma [10], Brian W. Davis [3], Teruaki Tozaki [10], Gabriella Lindgren [4] & Leif Andersson [1,3] ✉

The Greying with age phenotype in horses involves loss of hair pigmentation whereas skin pigmentation is not reduced, and a predisposition to melanoma. The causal mutation was initially reported as a duplication of a 4.6 kb intronic sequence in *Syntaxin 17*. The speed of greying varies considerably among Grey horses. Here we demonstrate the presence of two different *Grey* alleles, *G2* carrying two tandem copies of the duplicated sequence and *G3* carrying three. The latter is by far the most common allele, probably due to strong selection for the striking white phenotype. Our results reveal a remarkable dosage effect where the *G3* allele is associated with fast greying and high incidence of melanoma whereas *G2* is associated with slow greying and low incidence of melanoma. The copy number expansion transforms a weak enhancer to a strong melanocyte-specific enhancer that underlies hair greying (*G2* and *G3*) and a drastically elevated risk of melanoma (*G3* only). Our direct pedigree-based observation of the origin of a *G2* allele from a *G3* allele by copy number contraction demonstrates the dynamic evolution of this locus and provides the ultimate evidence for causality of the copy number variation of the 4.6 kb intronic sequence.

Greying with age in horses shows dominant inheritance and is one of the most iconic mutant phenotypes in animals[1]. These horses are born fully pigmented and usually start greying during their first year of life and most eventually become completely white. The beauty of these white horses has had a major impact on human culture, white horses occur abundantly in art, sagas, and fiction, and have most certainly been an inspiration for the myths regarding Pegasus and the Unicorn. An important reason for the popularity of this phenotype is that the

[1]Science for Life Laboratory, Department of Medical Biochemistry and Microbiology, Uppsala University, Uppsala, Sweden. [2]Institute of Marine Research, Bergen, Norway. [3]Department of Veterinary Integrative Biosciences, College of Veterinary Medicine and Biomedical Sciences, Texas A&M University, College Station, Texas, USA. [4]Department of Animal Sciences, Swedish University of Agricultural Sciences, Uppsala, Sweden. [5]Swedish Connemara Pony Breeders' Society, Falkenberg, Sweden. [6]Veterinary Genetics Laboratory, School of Veterinary Medicine, University of California-Davis, Davis, California, USA. [7]Department of Population Health and Reproduction, School of Veterinary Medicine, University of California, Davis, CA, USA. [8]Racehorse hospital, Miho Training Center, Japan Racing Association, Ibaraki, Japan. [9]Hidaka Training and Research Center, Japan Racing Association, Hokkaido, Japan. [10]Genetic Analysis Department, Laboratory of Racing Chemistry, Tochigi, Japan. [11]Present address: Childhood Cancer Research Unit, Department of Women's and Children's Health, Karolinska Institute, Tomtebodavägen 18A, 17177 Stockholm, Sweden. [12]These authors contributed equally: McKaela Hodge, Rakan Naboulsi. ✉e-mail: leif.andersson@imbim.uu.se

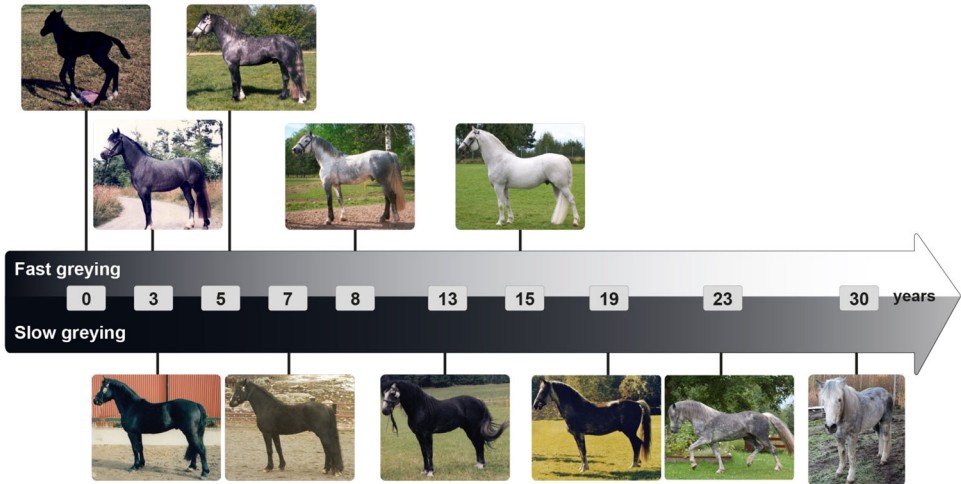

**Fig. 1 | Illustration of fast and slow greying Connemara Ponies.** Two individual horses, one fast greying (upper timeline) and one slow greying (lower timeline), were used for the illustration. Photo, from left to right: Fast Grey 1,2 Madeleine Beckman, 3 Maria Johansson, 4,5 Beckman Archive. Slow Grey 1–4 Maria Johansson, 5-6 Marlen Näslin.

causal mutation only affects hair pigmentation while skin and eye pigmentation are unaffected. Additionally, there are no noticeable negative effects on vision or hearing as is the case for many other pigmentation mutations in vertebrates with pleiotropic effects. For instance, mutations in the microphthalmia-associated transcription factor gene (*MITF*) are associated with deafness in dogs[2], Waardenburg syndrome type 2[3] (WS2A; MIM:193510) and Tietz albinism-deafness syndrome[4] (TADS; MIM:103500) in humans, and reduced eye size and early-onset deafness in mice[5]. Grey horses are believed to be as fit as any other horse, with the exception of the melanoma prevalence described below, and are as fast as other horses as illustrated by the fact that they successfully compete in horse races with horses of other colors. The mutation is widespread and is particularly common in some breeds, for instance Lipizzaner and Arabian horses. One negative side effect, though, is that in grey horses there is a high incidence of melanomas that occur in glabrous skin with increasing incidence by age[1] (Supplementary Fig. 1). The causal mutation for Grey was originally identified as a 4.6 kb duplication in intron 6 of *Syntaxin 17* (*STX17*)[1]. However, a recent study[6], based on digital PCR, indicated that the *Grey* haplotype carries three copies of the duplicated fragment. Horses homozygous for the *Grey* mutation (*G/G*) are reported to have significantly higher melanoma incidence than heterozygotes (*G/g*)[1]. The grey melanoma is usually benign, but can grow to sizes that affect quality of life and can lead to metastasis with fatal outcomes[7–9].

The *Grey* mutation causes upregulated expression of both *STX17* and the neighbouring gene *NR4A3*, encoding an orphan nuclear receptor[1]. Previous studies using both transfection experiments in a pigment cell line and transgenic zebrafish demonstrated that the duplicated region contains a melanocyte-specific enhancer[10]. The enhancer is evolutionarily conserved and contains two binding sites for microphthalmia transcription factor (MITF), a master regulator of gene expression in pigment cells[5]. MITF knock-down silenced pigment cell-specific expression from a reporter construct with the horse copy number variation in transgenic zebrafish[10]. It is still an open question whether upregulation of *STX17* or *NR4A3* or their combined effect is critical for the phenotypic effects of the *Grey* mutation. No other study has revealed a direct link between these genes and pigment biology but both have links to cancer. The STX17 protein has a key role in autophagy[11], a pathway currently explored as target for melanoma therapy in humans[12]. The function of NR4A3 as a regulator of transcription is only partially understood, but data indicate that this

orphan nuclear receptor has a role in maintaining cellular homeostasis and in pathophysiology including tumour development[13].

It is well known that the speed of greying varies considerably among Grey horses. One important factor is the *Grey* genotype as *Grey* homozygotes grey faster than *Grey* heterozygotes[1]. However, the variation in the speed of greying is pronounced in certain breeds such as Connemara ponies. In this breed two distinct types of greying are noted, fast greying and slow greying (Fig. 1). Fast greying horses are usually completely white at about 10 years of age whereas the slow greying horses never become completely white but show a beautiful dappled grey color at older ages. The aim of the present study was to explore the genetic basis for this striking phenotypic difference.

Here we report that speed of greying and melanoma incidence are controlled by the *Grey* locus itself rather than by other genes. The causal difference is copy number variation (CNV) of the duplicated sequence, fast greying horses carry at least one allele with three copies of the 4.6 kb sequence while slow greying horses carry an allele with only two copies. Thus, the *Grey* locus harbours an allelic series with at least three alleles: *G1* – one copy, wild type; *G2* – two copies, slow greying; *G3* – three copies, fast greying (Fig. 2).

## Results
### Speed of greying maps to the *Grey* locus
We identified a family of Connemara ponies apparently segregating for fast and slow greying (Supplementary Fig. 2; Supplementary Table 1). We took advantage of this pedigree in an attempt to identify the genetic basis for the difference in speed of greying. The key individual was a fast greying sire Hagens D'Arcy (stud book number RC 101) in generation 5 whose mother was fast greying and whose father greyed slowly. Among the 16 progeny from matings to non-grey (*G1/G1*) dams, 10 were classified as slow greying while 6 were fast greying. This suggested that the speed of greying in this family shows Mendelian inheritance and the observed proportion of the two types did not differ significantly from an expected 1:1 ratio if the sire is heterozygous for alleles associated with fast and slow greying ($X^2 = 1.0$, d.f. = 1; $P > 0.05$). We also had access to a small family of Thoroughbred horses from Japan comprising two horses classified as fast greying and three as slow greying (Supplementary Fig. 3).

The putative locus controlling speed of greying may be due to sequence variation at the *STX17/Grey* locus itself, a linked locus, or an unlinked locus. To distinguish these possibilities, we carried out whole

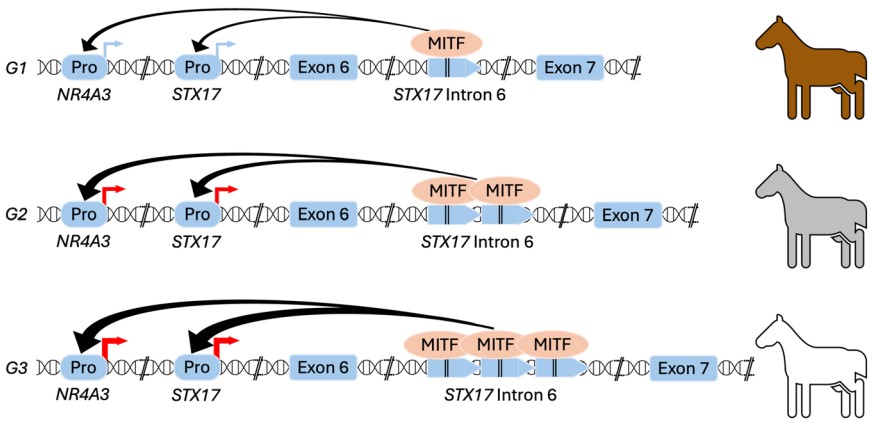

**Fig. 2 | Schematic illustration of the nature and significance of the *Grey* mutation.** The causal mutation for the Grey phenotype is a copy number variation of a 4.6 kb sequence located in *STX17* intron 6. The duplicated sequence harbours two binding sites for MITF and the presence of copy number expansion upregulates the expression of both *NR4A3* and *STX17* in cis[1]. *G1* is the wild-type allele with a single copy of the 4.6 kb sequence, *G2* carries two copies and causes slow greying, and *G3* carries three copies and causes fast greying. Pro promoter.

genome sequencing of a total of 13 slow greying and 9 fast greying horses, including the Connemara pony sire Hagens D'Arcy. A genome-wide association analysis, using all sequenced horses except the sire Hagens D'Arcy (Supplementary Table 2) and individual SNPs, revealed a single consistent signal of association on chromosome 25 harbouring the *STX17/Grey* locus, with six SNPs showing perfect association to speed of greying (Fig. 3a), none of which were located in coding or conserved sequences (Supplementary Table 3). Further characterization of the signal of association revealed that it was centered around the *STX17/Grey* locus (Fig. 3b). This signal of association does not reach genome-wide significance since the *P*-value (Fisher's exact test) for the most strongly associated SNPs are $4.9 \times 10^{-6}$ and we have used 8,426,579 SNPs in the analysis. However, the fact that 394 out of the 397 most strongly associated SNPs map within 10 Mb of *STX17* provides compelling support for this location.

We next hypothesized that the causal difference between fast and slow greying could be a copy number variation of the 4.6 kb duplicated sequence in *STX17*. We determined the copy number both by sequence coverage in the whole genome sequencing data and by droplet digital PCR (ddPCR) of the duplicated sequence first using samples from the Connemara pony family (Supplementary Table 2). This analysis showed that the fast greying sire Hagens D'Arcy carried 5 copies of the duplicated sequence (deduced to be 3 + 2 copies on the two chromosomes), the fast greying progeny carried 4 copies (3 + 1 because their dams were wild-type and transmit an allele with a single copy), and the slow greying progeny carried 3 copies (2 + 1). Analysis of sequence coverage and ddPCR data obtained for the five Japanese Thoroughbred horses confirmed the same pattern. The perfect association between copy number of the duplicated sequence and the fast/slow greying phenotype including both Connemara ponies from Sweden and Thoroughbreds from Japan is highly significant (Fisher's exact test; $P = 4.9 \times 10^{-6}$) (Fig. 3c, Supplementary Fig. 4a, Supplementary Table 2). Based on these results we propose a new nomenclature for the *Grey* locus: *G1* - one copy, wild type, previously reported as N by most genetic testing laboratories; *G2* – two copies, slow greying; *G3* – three copies, fast greying (Fig. 2). The fact that the *G2* allele has not been found among the many hundreds, if not thousands, of non-grey horses tested for the presence of the 4.6 kb duplication provides conclusive evidence that slow greying is determined by an allele at the *Grey* locus.

The great majority of, if not all, Grey horses previously studied and determined to be fast greying appear to carry *Grey* haplotypes that share a 350 kb region that is identical-by-descent (IBD)[1,14]. This implies that the initial mutation can be traced back to a single mutational event

that most likely happened subsequent to horse domestication. An important question is therefore whether the two-copy *Grey* haplotype shares the same IBD region as the three-copy haplotype. IBD sharing between *G2* and *G3* is supported by the Connemara sire Hagens D'Arcy, which is heterozygous for the two haplotypes and show essentially no sequence diversity in the 350 kb IBD region (Supplementary Fig. 4b). Pedigree analysis identified a Connemara pony as a putative *G2/G2* homozygote and this genotype was confirmed by Nanopore (ONT) adaptive sampling, long-read sequencing. Sequence analysis of the entire chromosome 25 revealed that this horse showed runs of homozygosity over a 16 Mb region including the *Grey* locus (chr25:2-18 Mb; Supplementary Fig. 5a), suggesting that both *G2* haplotypes trace back to a relatively recent common ancestor.

The sequencing of this *G2/G2* homozygote allowed us to establish a *G2* haplotype sequence across the *STX17/Grey* region and in several Mb of the flanking regions. We took advantage of this information and identified sequence differences between this haplotype and all other *G2* and *G3* haplotypes sequenced in this study. Sequence differences were inferred when the *G2/G2* homozygote and another horse were homozygous for different SNPs (Fig. 3d). This analysis provided three important results. First, all *G2* and *G3* haplotypes share the previously described 350 kb IBD region consistent with a common origin of the two haplotypes. Second, all *G2* haplotypes from Connemara ponies are IBD for at least a 4.5 Mb region including *STX17/Grey*. Three, sequence identities between *G2* haplotypes from Connemara ponies and Japanese Thoroughbred horses break down outside the 350 kb IBD region shared by all *G2* and *G3* haplotypes (Fig. 3d), implying that they do not have a recent common ancestry.

**Targeted long read sequencing supports the causal nature of the copy number variation**

Nanopore (ONT) Cas9-targeted sequencing (nCATS) makes it possible to phase chromosome fragments of tens of kb without amplification and thereby establishing phase, structure and methylation status of individual haplotypes, including the sequence orientation of duplicated fragments[15]. We used this approach to characterize the two grey haplotypes of Hagens D'Arcy from the Connemara pony family; the genotype of this sire is *G2/G3*. ONT sequencing of DNA fragments spanning the duplicated region including about 2 kb of flanking sequences on each side confirmed that the sire is heterozygous *G2/G3* based on the detection of two dominant fragments sizes: a 13 kb sequence comprising 2 copies of the duplicated sequence and a 17 kb sequence comprising 3 copies (Fig. 4a). After generating separate consensus sequences of reads corresponding to the *G2* and *G3* alleles,

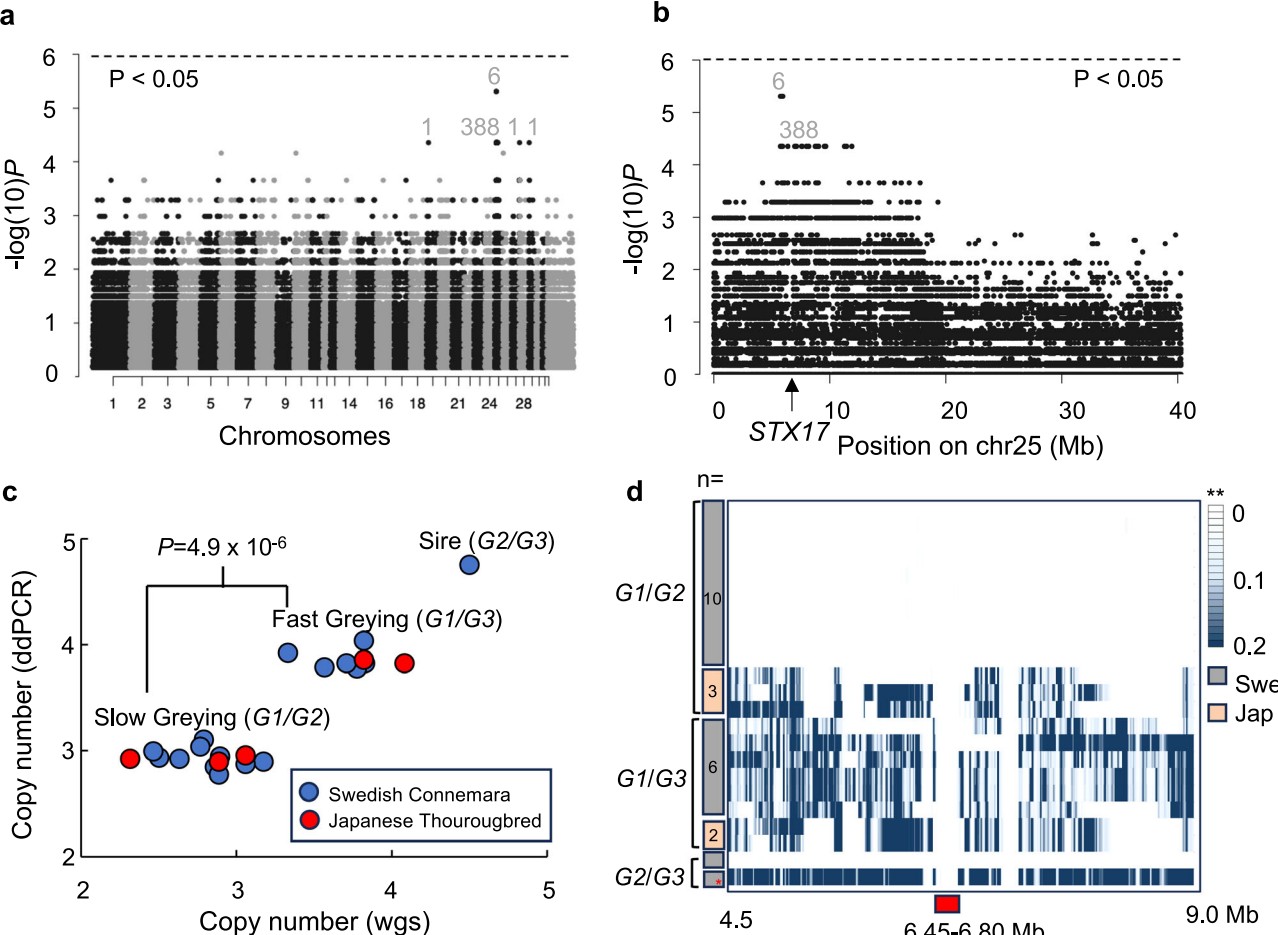

**Fig. 3 | Whole genome sequencing reveals copy number difference at the *STX17*/ *Grey* locus between fast and slow greying horses. a** GWAS contrasting fast and slow greying horses belonging to the Connemara pony breed and Japanese Thoroughbred horses. The number of the most highly associated SNPs ($P \le 4.4 \times 10^{-5}$) at different genomic positions are indicated. The Bonferroni-corrected genome-wide significance threshold is indicated (see methods). **b** GWAS zoom-in on chromosome 25. The location of *STX17* is marked with an arrow. Numbers of SNPs are written out above the most highly associated loci. **c** Estimated *STX17* copy number variation in slow greying and fast greying horses from the Swedish Connemara and Japanese Thoroughbred pedigrees. **d** Evaluation of IBD intervals between *G2* and *G3* haplotypes. **\*\***Heat map colors indicate fractions of opposite homozygote calls, in sliding windows of 100 SNPs, between a Connemara *G2/G2* homozygote compared with 10 + 3 slow greying (*G1/G2*) and 6 + 2 fast greying (*G1/G3*) horses from the Swedish Connemara + Japanese Thoroughbred pedigrees. The comparison with the Connemara sire Hagens D'Arcy (*G2/G3*) includes both the fraction of homozygous (upper) and heterozygous differences (lower) revealing the proportion of differences to the *G2* and *G3* haplotypes, respectively. The IBD region (chr25:6.45-6.80 Mb) is highlighted by a red box. Significance values indicated in (**a–c**) are based on Fisher's exact test, two-sided, not corrected for multiple testing.

alignment of these sequences to the horse reference genome assembly (EquCab3) shows that the duplicated copies both in *G2* and in *G3* are located in tandem and are oriented head-to-tail (Fig. 4b). Characterization of the CNV breakpoint sequences revealed a six bp microhomology motif (TCTCAG) immediately adjacent to the breakpoints on each side (Supplementary Fig. 6). It is known that, following a double strand break, sequence duplication can be initiated by microhomology-mediated break-induced replication (MMBIR)[16]. Thus, it is possible that the duplication initially arose by this or other mechanisms involving this motif. Once duplicated, further copy number expansions/retractions of the CNV may involve DNA double-strand break repair pathways such as Non-Allelic Homologous Replication (NAHR) or single strand annealing[16].

ONT sequencing makes it possible to also analyze the DNA methylation pattern of the sequence. Interestingly, these data revealed that the evolutionary conserved region harbouring MITF binding sites[10] shows a clear reduction in DNA methylation using genomic DNA isolated from blood (Fig. 4c), a result supporting our previous interpretation that this region acts as an enhancer and that its copy number variation is causal for the phenotypic effects[10]. Oxford nanopore

sequencing based analysis of blood CpG methylation in the *G2/G2* homozygote horse and two *G1/G3* horses verified low DNA methylation in the same region (Supplementary Fig. 7).

### The birth of a *G2* allele
The fact that the Japanese Thoroughbred family included in this study constitutes the only reported slow greying horses in this breed prompted us to explore the possibility that the *G2* allele may have arosen by mutation in one of the recent ancestors of the family. We used ddPCR to genotype all ancestors, for which genomic DNA was available, of the first slow greying horses noted in this family: the slow greying individual X in generation 2 highlighted in Fig. 5 and depicted in Supplementary Fig. 3. This analysis showed that the *G2* allele must have originated by mutation during meiosis in the dam of individual X because the dam's genotype was *G3/G3* while the sire was non-grey (*G1/G1*) and transmitted the *G1* allele to individual X.

We next performed whole genome sequencing of the dam and sire of individual X in an attempt to determine whether the *G2* allele arose by an intrachromosomal or interchromosomal (unequal crossing-over) event. This bioinformatic analysis confirmed that the

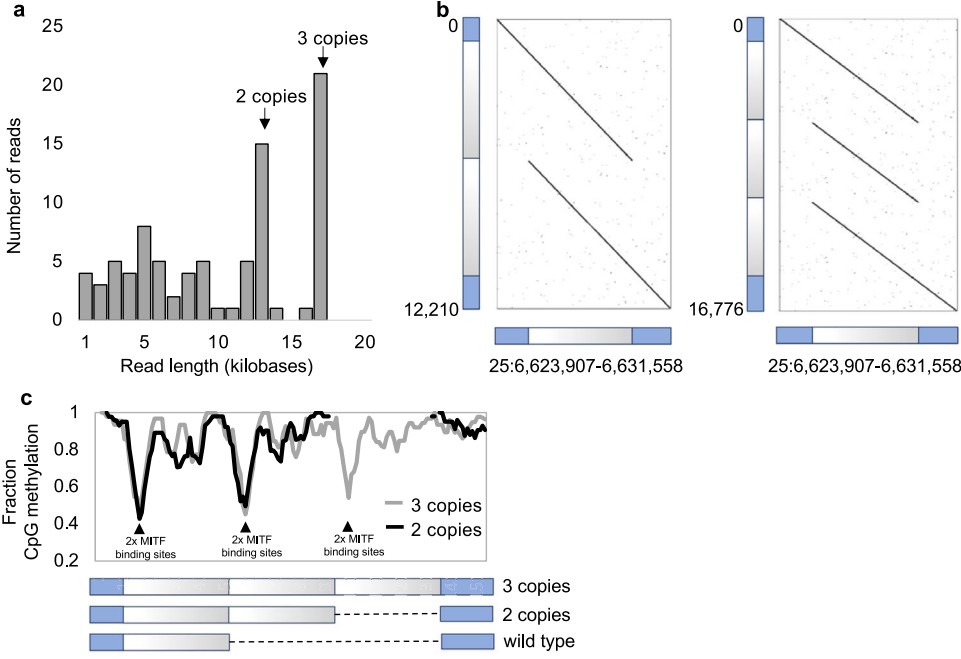

**Fig. 4 | Copy number variation and DNA methylation patterns revealed by sequence capture and Nanopore sequencing. a** Molecular size distribution of the *STX17* region from a *G2/G3* heterozygote (Hagens D'Arcy) inferred from a Cas9 capture Nanopore sequencing experiment. **b** The duplicated sequences are present in tandem both on the *G2* and *G3* alleles. The nucleotide positions in bp in the genome assembly and in the Nanopore reads are shown on the x- and y-axis, respectively. **c** DNA methylation pattern associated with the *G2* (black) and *G3* (grey) alleles. The drastic reduction of DNA methylation at precisely the region harbouring MITF binding sites are high-lighted.

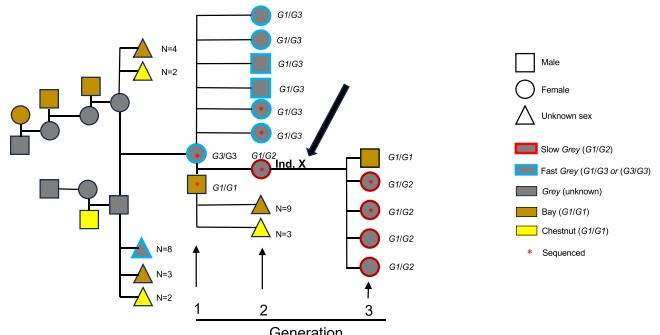

**Fig. 5 | The birth of a *G2* allele in a pedigree of Japanese Thoroughbred horses segregating for the slow greying allele.** The coat color and the *Grey* genotype determined with ddPCR (when available) are indicated. Individual X in generation 2 is the same as individual X depicted in Supplementary Fig. 3.

dam was homozygous *G3/G3* and mother of individual X. Unfortunately, she showed complete homozygosity for the interval 0-11 Mb on chromosome 25, including the *Grey* locus at 6.6 Mb (Supplementary Fig. 8). Thus, it is impossible to determine whether the mutation was caused by unequal crossing-over or another mechanism.

### Genotype distribution across breeds

There is an interest from horse owners and breeders to determine the genotype at the *Grey* locus in horses. During the last 6 years the Veterinary Genetics Laboratory (UC Davis) has utilized a ddPCR assay to test for *Grey* genotypes. Including data from those samples whose owner or registry consented to be included in research allowed us to evaluate 1400 horses representing 78 breeds/populations. These data enabled the determination of the grey copy number distribution across breeds. The majority of samples tested had four copies ($n = 831$). It is not possible to know from this assay alone if these animals are homozygous *G2/G2* or heterozygous *G1/G3*. However, despite non-random sampling, in those breeds where we did not identify a single *G1/G2* heterozygote it is extremely unlikely that all individuals with four copies are homozygous *G2/G2*. Under this assumption, the *G3* allele was identified in 62 different populations (Supplementary Data 1 and 2). Whereas the *G2* allele was only definitively detected in 8 of these 62 populations (horses with 3 and/or 5 copies detected, Table 1). Under the assumption that the genotype distribution at this locus does not show an extreme deviation from Hardy-Weinberg equilibrium, the *G2* allele was always much rarer than *G3* in the population where evidence of both alleles occurred, because horses with 4 copies were always more abundant than those with three copies in the ddPCR screens. We did not find evidence for the presence of an allele with 4 or more copies which would be evident if a horse has 7 or more copies in total. Our data indicate that if such alleles exist, they must be very rare.

### The slow greying phenotype is not associated with a high incidence of melanoma

Our observation that the *G2* allele has a much milder effect on the greying process than the *G3* allele begs the question whether it is also associated with a lower incidence of melanoma. We identified 25 late grey (*G1/G2*) Connemara horses, 15 years of age or older (range 15–35 years) and searched for the presence of the classical grey melanomas present on glabrous skin (lips, eye lids, under the tail) but detected none among these horses (Supplementary Table 4). Previous studies have documented that from 50 – 80% of all Grey horses of this age carry melanoma[7–9]. This dramatic difference in incidence 80% vs. 0% in 25 slow-greying horses is highly significant ($X^2 = 25.0$, d.f. = 1; $P = 6 \times 10^{-7}$). This study involved a single confirmed *G2/G2* homozygote from the Connemara breed. The coat color phenotype of this single horse was classified at the age of 11 years as intermediate between a

**Table 1 | Copy number variation for the 4.6 kb duplicated sequence in *STX17* intron 6 in populations in which the *G2* and *G3* alleles were detected**

| Breed | Copy number (deduced genotype) | | | | | | Allele frequency[a] | |
|---|---|---|---|---|---|---|---|---|
| | 2 (G1/G1) | 3 (G1/G2) | 4 (G1/G3 or G2/G2) | 5 (G2/G3) | 6 (G3/G3) | n | G2 | G3 |
| Andalusian | 25 | 8 | 40 | 1 | 9 | 83 | 0.05 | 0.36 |
| Connemara Pony | 23 | 6 | 49 | 3 | 9 | 90 | 0.05 | 0.39 |
| Miniature Horse | 25 | 1 | 15 | 0 | 0 | 41 | 0.01 | 0.18 |
| Mangalarga Marchador | 0 | 1 | 8 | 0 | 0 | 9 | 0.06 | 0.44 |
| Mustang | 3 | 2 | 8 | 0 | 0 | 13 | 0.08 | 0.31 |
| Quarter Horse | 141 | 0 | 284 | 1 | 13 | 439 | 0.00 | 0.35 |
| Tennessee Walking Horse | 16 | 2 | 23 | 0 | 4 | 45 | 0.02 | 0.34 |
| Welsh Pony | 7 | 1 | 25 | 0 | 3 | 36 | 0.01 | 0.43 |

[a]Allele frequency calculated under the assumption that no *G2/G2* homozygote is present among these samples due to the low expected frequency determined by the number of *G1/G2* and/or *G2/G3* heterozygotes. These are biased estimates of allele frequencies in these breeds but give an indication of the relative frequency of the *G2* and *G3* alleles.
Data based on genotyping services at the UC Davis Veterinary Genetics Laboratory.

slow greying (*G1/G2*) and a fast greying horse (*G1/G3*) (Supplementary Fig. 5b) and it had no visible melanoma.

## Discussion

This study provides conclusive evidence for the presence of two variant alleles at the *Grey* locus in horses in addition to the *G1* wild-type allele. The most common allele, *G3*, carries three tandem copies of the 4.6 kb sequence and is associated with fast greying that eventually results in a white coat color, and the allele is associated with a high incidence of melanoma. In contrast, the *G2* allele carries a tandem duplication and *G1/G2* heterozygotes grey slowly, never become pure white, and have a significantly lower incidence of melanoma compared with *G1/G3* heterozygotes, possibly as low as *G1/G1* homozygotes. The fact that *G2* and *G3* share a 350 kb IBD region implies that the two alleles originate from the same ancestral mutation, but it is unknown if the ancestral allele carried two or three copies of the 4.6 kb sequence. Our finding of the origin of a *G2* allele by mutation in a *G3/G3* parent demonstrates the dynamic evolution at this locus in which alleles can be generated from each other by unequal crossing-over or slippage during DNA replication. How common such copy number contractions/expansions are at the *Grey* locus is unknown. We found a copy number contraction in one out of two pedigrees, but that particular pedigree was included in this study because of the sudden appearance of an unexpected phenotype in Japanese Thoroughbred horses.

Our across breed analysis shows that *G3* is much more common than *G2*. This is most likely caused by strong selection for the splendid white phenotype associated with this allele. Our results now solve the mystery why some grey horses never become pure white. Although this study explains a considerable portion of the heterogeneity in speed of greying, it is likely that additional genetic variation elsewhere in the genome contributes to variation in the speed of greying and/or contribute to melanoma risk. It has been shown that in addition to the large effect of the *STX17* mutation, polygenic inheritance contributes to the speed of greying and incidence of melanoma in Lipizzaner horses[17] and *DPF3* has been reported as a putative candidate gene affecting the incidence of grey horse melanoma[18]. It is also possible that further genetic heterogeneity exists at the *Grey* locus itself.

There is a remarkably strong dose sensitivity at the *Grey* locus. The *G1/G1* genotype shows no greying and low incidence of melanoma, *G1/G2* shows slow greying and low incidence of melanoma, *G1/G3* shows fast greying and high incidence of melanoma, and *G3/G3* homozygotes grey very fast and show a very high incidence of melanoma as previously demonstrated[1]. There are too few observations to make firm predictions of the phenotype of *G2/G3* heterozygotes and *G2/G2* homozygotes. However, it is likely that *G2/G3* horses are intermediate compared with the *G1/G3* and *G3/G3* genotypes with regard to hair

greying and melanoma incidence. The single *G2/G2* individual confirmed by sequence capture and Nanopore sequencing showed a grey phenotype and no melanoma at the age of 11 years (Supplementary Fig. 5) whereas a *G1/G3* heterozygote is expected to have developed white color and is likely to have some melanoma at the same age. This suggests that having a triplication on one chromosome gives a stronger transcriptional activation than having a duplication on both chromosomes. The present study has important practical implications for horse breeding because it suggests that it is possible to have a horse showing the beauty of the grey coat color (Fig. 1; Supplementary Fig. 5) most likely with no elevated risk to develop melanoma. It also illustrates the need for additional testing methodologies to determine zygosity of this tandem repeat in those horses with four copies (*G2/G2* or *G1/G3*). Current testing methodologies, which do not routinely use long read sequencing for many reasons, are therefore not yet able to resolve this issue.

Our results provide strong support for the causality of the CNV on the Grey phenotype, because of the remarkable correlation between the copy number of the 4.6 kb sequence and phenotype. Furthermore, our direct pedigree-based observation of a mutation from *G3* to *G2* and the associated shift in phenotype provides evidence for causality (Fig. 5; Supplementary Fig. 3). The results also supports the interpretation that the triplication underlies both loss of hair pigmentation through its effect on hair follicle melanocytes and melanoma incidence through its effect on dermal and/or epidermal melanocytes. The higher copy number enhances the effect on both speed of greying and melanoma risk. Furthermore, *G2* alleles from two different breeds (Connemara pony and Thoroughbreds) share a 350 kb IBD region with *G3* alleles from many different breeds making the copy number of the duplicated sequence the distinctive difference between alleles. The 4.6 kb sequence showing copy number variation contains two binding sites for MITF, a master regulator for transcription in pigment cells[5]. A previous study based on both cellular transfection experiments and transgenic zebrafish indicated that this copy number expansion transforms a weak enhancer to a strong melanocyte-specific enhancer[10]. The drastic drop in DNA methylation precisely at the region where the MITF binding sites are located (Fig. 4c, Supplementary Fig. 7) despite the fact that we used genomic DNA from blood cells for Nanopore sequencing is consistent with the notion that the 4.6 kb sequence contains an element affecting gene regulation. It is likely that the *STX17* CNV is a direct driver of melanoma development because we noted further copy number expansion in melanoma cells in horses, up to nine copies in tumour tissue[14].

A classical feature of tandem duplications is expansions and contractions of the copy number in the germ line due to non-allelic homologous recombination[19]. This is well illustrated by our finding of a

*G3* to *G2* mutation in the Japanese Thoroughbred pedigree. Thus, since we observe haplotypes with both two and three copies it is expected that haplotypes with four or more germ-line copies may occur. Genotyping 1400 horses representing 78 populations has not revealed further expansion of this tandem repeat beyond three copies in the germ line. This observation together with our finding of low DNA methylation of the enhancer region also in blood cells and the strong dosage effect suggest that four or more tandem copies of the duplicated sequence may be deleterious and thus exist at very low frequency, if at all.

## Methods

### Horse samples

All horses included in this study were privately owned.

The Connemara pedigree from Sweden consisted of the sire Hagens D'Arcy and 16 of his offspring from matings to non-grey mares. Blood sample collection was approved by the ethics committee for animal experiments in Uppsala, Sweden (number: 5.8.18-15453/2017 and 5.8.18-01654/2020). Genomic DNA was extracted on the Qiasymphony instrument with the Qiasymphony DSP DNA mini or midi kit (Qiagen, Hilden, Germany).

The Japanese Thoroughbred pedigree comprised multiple generations (Fig. 5; Supplementary Fig. 3). Blood collections were approved by the Animal Care Committee of the Laboratory of Racing Chemistry (Utsunomiya, Tochigi, Japan, Approval Number: 20-4) and horsehair roots were provided from the Japan Association for International Racing and Stud Book (JAIRS) under their permission. All thoroughbred racehorses in Japan are registered by the JAIRS following parentage determination tests. DNA extraction from whole blood was performed using a DNeasy Blood & tissue Kit (Qiagen, Hilden, Germany). DNA extraction from horsehair roots was performed using a MagExtract genome kit (Toyobo, Osaka, Japan) with minor modifications[20].

Blood samples from two fast greying Quarter horses were collected for the purpose of Nanopore Cas9-targeted sequencing under ethical permit by the Institutional Animal Care & Use Committee (IACUC) 2017-0152 and IACUC 2023-0267 (Texas A&M University). Both horses were extensively grey with fleabitten coloration, had mild vitiligo, and had a history of multiple grey horse melanomas.

### Grey phenotyping

All Grey horses, fast or slow greying, are born with a primary fully pigmented coat, but the onset and progress of greying differ markedly between slow and fast greying horses. A fast greying foal will normally display a few grey hairs on the eyelids during the first week after birth (Supplementary Fig. 9a). During the first month more grey hairs appear as grey "circles" around the eyes (Supplementary Fig. 9b), grey eyelashes and grey hairs on the tail root develop too. In contrast, a slow greying foal has no visible grey hairs on the eyelids at birth and does not develop grey "eye circles" or any grey hairs elsewhere during its first month of life. If a slow greying foal has face markings (like a star or blaze) the white parts are not well-defined and the white hairs will spread out in time (Supplementary Fig. 9c). The first signs of greying in a slow greying horse will normally not appear until 5–7 years of age, with grey hairs often expanding from white head markings (Supplementary Fig. 9d). Grey eyelashes can appear when the head is still dark colored with no grey hairs on the eyelids. Grey hairs may appear partly on the neck. The next phase of greying will include head, neck, distal parts of the leg and the tail. When a slow greying horse has reached an age of about 15 years the coat color may turn into a dappled steel or bluish grey shade. Although there is individual variation in the speed of greying also among slow greying horses, they will never become a visually white horse (Fig. 1); Supplementary Fig. 9e–g show the slow greying process of the same horse at different ages, but without head markings. The breeder of the slow greying *G2/G2* homozygote from the Connemara pony in our study stated that the horse had developed the slow greying color like a normal slow Grey Connemara pony, but that it had turned grey earlier than expected, but not at all as early as a fast greying heterozygote (*G1/G3*). In contrast to heterozygous slow greying horses (*G1/G2*), it started with greying of the head much like a fast greying horse. As a 3-year-old, it began to have grey hair on its head and it lacked white eyelashes. At 8 years old, it had a light head but the body was still dark. Only when it was 11 years old the grey color developed on the body and the characteristic dappled colored pattern began to appear on the front of the horse (Supplementary Fig. 5a). The head is still lighter but overall, the horse differs markedly from a fast-greying horse of the same age.

### Whole genome sequencing

A Covaris E220 system was used to fragment 1 μg of genomic DNA to a target insert size of 350-400 bp. Sequencing libraries of individual DNA samples were prepared using the Illumina TruSeq DNA PCR-free LP kit (Illumina, San Diego, California, United States) in combination with the IDT for Illumina TruSeq DNA UD indexes kit (Illumina). Preparation of the libraries was performed according to the manufacturer's protocol. A TapeStation with the D1000 ScreenTape (Agilent Technologies, Santa Clara, California, USA) was used to assess the quality of the libraries. qPCR was performed using the Library quantification kit for Illumina (KAPA Biosystems, Basel, Switzerland) on a CFX384 Touch instrument (Bio-Rad, Hercules, California, USA) to quantify the adaptor-ligated fragments prior to cluster generation and sequencing. Indexed samples were sequenced on a NovaSeq S4 flow cell. Genomic DNA from the Japanese Thoroughbred horses was purified using the DNeasy Blood & Tissue Kit (Qiagen, Hilden, Germany) from whole bloods collected in BD Vacutainer K2E Plus Blood Collection Tubes (Becton, Dickinson and Company, Franklin Lakes, NJ, USA). Sequencing was carried out by Macrogen Japan Corp. (Koto, Tokyo, Japan) using the TruSeq DNA PCR-Free Library Prep Kit and NovaSeq 6000 platforms (Illumina, San Diego, CA, USA).

The demultiplexed sequences were aligned to the horse genome (EquCab3) using bwa-mem2 2.0pre2[21]. Read duplicates were marked using MarkDuplicates Picard Tools, version 3.1.1 (http://broadinstitute.github.io/picard) and resulting bam-files were used for variant calling by UnifiedGenotyper (GATK version 3.8–0)[22]. Depth of coverage was determined by means of the GATK module DepthOfCoverage and the resulting per-base coverage files were used to calculate mean depths of coverage across samples for windows of varying sizes. In order to estimate DNA copy numbers in windows, depths of coverage in individual windows were normalized to the genome average depth. Kinship coefficients and nucleotide diversity estimates were obtained using the relatedness2 and window-pi functions of vcftools[23] version 0.1.16, respectively.

### Genome Wide Association (GWAS) analysis

Genotype data in vcf format was converted to PLINK format using vcftools[23]. GWAS was performed in PLINK[24] version v1.90b7.2 using the "--assoc" function with the Fisher's exact test and a dominance model, with phenotypes of 8 fast greying individuals encoded as "1" and phenotypes of 13 slow greying individuals encoded as "2". Resulting P-values were imported into R[25] (distribution 4.3.1) and were plotted along the genome using the package qqman[26] (version 0.1.9). We caculated a genome-wide, Bonferroni-corrected significance treshold as $0.05/50 \times 10^3 = 1 \times 10^{-6}$ based on an experimental design where we would select 50,000 highly informative SNPs (minor allele frequency > 0.3) evenly spread across the genome (about one SNP per 50 kb in the horse genome) from the 8,426,579 SNPs detected using whole genome resequencing. This experimental design was appropriate for mapping the speed of greying locus because pedigree data was consistent with a monogenic inheritance.

### Nanopore Cas9-targeted sequencing of a *G2/G3* individual and two *G1/G3* individuals

High molecular weight DNA was isolated from blood of the *G2/G3* Connemara sire (Hagens D'Arcy). High molecular weight DNA from the two fast greying Quarter horses was extracted from blood by the Texas A&M Institute for Genome Science and Society (TIGSS) Experimental Genomics Core. We used a Cas9 sequence capture protocol (Ligation sequencing gDNA−Cas9 enrichment; SQK-CS9109) to construct a sequencing library where DNA molecules spanning the 4.6 kb duplication/triplication were much more likely than other fragments to ligate with the sequencing adaptor. Sequencing was performed on a R9.4.1 flow cell using a MinION instrument and the Cas9 Sequencing Kit (SQK-CS9109; Oxford Nanopore Technologies). We designed four guideRNAs (Supplementary Table 5) and purchased these as *S. pyogenes* Cas9 Alt-R crRNAs (Integrated DNA Technologies). Alt-R *S. pyogenes* Cas9 tracrRNA (Integrated DNA Technologies) and Alt-R *S. pyogenes* HiFi Cas9 nuclease V3 (Integrated DNA Technologies) were used to cut DNA. Base calling was performed using Guppy Version 6.5.7 (Oxford Nanopore Technologies) using the super accuracy setting. A wildtype haplotype with only one copy of the 4.6 kb sequence will result in a ~7.7 kb sequence while a haplotype with two copies will produce a ~12.3 kb sequence and a triplication will produce a ~16.9 kb sequence. Reads spanning the entire CNV region, i.e. those ascertained to carry either copy of the duplication, were separately binned and consensus sequences were determined for each of these. For each horse, binned reads were then aligned to their respective consensus sequence (*G1*, *G2* or *G3*) using minimap2[27] (version 2.26-r1175) and the resulting bam files and read fast5 files were used to call CpG methylation in nanopolish[28] (version 0.14.1) following the pipeline (https://nanopolish.readthedocs.io/en/latest/quickstart_call_methylation.html). For visualization, the per-site 5mC methylation fractions were averaged in sliding windows of 5 adjacent CpGs, moving forward one CpG site at a time.

### Nanopore sequencing by adaptive sampling of a *G2/G2* individual

High molecular weight DNA was isolated from blood of one individual predicted to be homozygous *G2/G2* based on pedigree analysis. This DNA was fragmented to 20 kb using a g-TUBE (Covaris) and molecules below ~10 kb in size were then depleted using the SRE XS kit (Pacific Biosciences of California). For sequencing library construction, the kit SQK-LSK114 was used and 40 fmol of the finished library was loaded onto a PromethION R10 flow cell (Oxford Nanopore Technologies) for sequencing. The run was started from within MinKNOW on a P2 instrument with adaptive sampling enabled to achieve high coverage on chr25. Adaptive sampling used the horse reference genome fasta file (EquCab3) and an accompanying bed-file specifying chromosomes 1–24 and chromosomes 26–27 (88% of the genome) as off-target coordinates. The resulting data in pod5 format was base called using dorado v0.5.1+a7fb3e3 (https://github.com/nanoporetech/dorado) using the superior (sup) model, simultaneously calling CpG methylation (5mC and 5hmC). Reads were aligned to the *G2* consensus sequence using minimap2[27] (version 2.26-r1175). Methylation status was assessed at 41 CpG sites for each copy of the duplication and at 15 and 17 CpG sites flanking the duplicated fragment up-and downstream, respectively. Per-site observed 5mC and 5hmC calls along the *G2* consensus sequence were merged to generate per-site total CpG methylation fractions. For visualization, total per-site methylation fractions were averaged in sliding windows of 5 adjacent CpGs, moving forward one CpG site at a time. The Nanopore reads were also used to estimate nucleotide diversity along chr25. Reads were aligned to equCab3 using minimap2[27] (version 2.26-r1175) and the resulting alignments were subjected to SAMtools[29] mpileup (version 1.20), using only alignments with a mapping quality of at least 20. The average depth of coverage of the sequenced *G2/G2* individual was

approximately 45x for chromosome 25 harbouring the *Grey* locus and to compensate for noise due to Nanopore basecalling errors, positions in the mpileup file with ≤2 reads supporting a non-major allele were set to a count of 0 non-major alleles. From the filtered mpileup file, per-site nucleotide diversity was then estimated by determining the fraction of non-major alleles observed. Averages of the per-site nucleotide diversity estimates were determined for 500 kb windows along chr25.

### Digital PCR and copy number analysis

Horses from the Swedish Connemara pony pedigree and controls were genotyped using two TaqMan assays designed to specifically target the *STX17* copy number variation (ECA25: 6625371-6625493 (NC_009168.3), Bio-Rad Assay dCNS718018816; and ECA25: 6629062-6629184 (NC_009168.3), Bio-Rad Assay dCNS933010968) and a third control assay to target a neighbouring region free of known CNVs (ECA25: 6624137-6624259 (NC_009168.3), Bio-Rad Assay dCNS851288843. A fourth assay targeting *myostatin (MSTN)*[30] was used as a reference in the ddPCR experiment. The assays were designed using EquCab3 as a reference genome. The ddPCR experiment was performed using the Bio-Rad QX200 Droplet Reader platform. Briefly, droplets were generated using the Bio-Rad Automated Droplet Generator instrument before placing the ddPCR mix in a thermal cycler where the PCR reaction was performed. Finally, the results were generated using Bio-Rad QX200 Droplet Reader platform. The ddPCR mix contained 11 μL of 2X ddPCR supermix for probes, 1.1 μL of target 20X TaqMan assays, 1.1 μL of reference 20X TaqMan assays, 1 μL of 20 ng/μL genomic DNA, 2 μL fast digest EcoRI, and water up to 22 μL. The mix was loaded into a droplet generator cartridge. Droplets were generated following the manufacturer's protocol. The PCR plate containing the droplets was foil sealed before the PCR reaction. The PCR setup was 95 °C for 10 min, 40 cycles of 30 s at 94 °C and 60 s at 60 °C. Then finally, 10 min at 98 °C. The PCR plate was transferred to the droplet reader for analysis. Cluster classification was manually adjusted in non-clear events. ddPCR analysis using the same method was also used to genotype horses from the Japanese Thoroughbred pedigree. Additionally, copy number variation at the Grey locus was evaluated for 1,400 horses across 78 breeds based on genotyping services provided by the UC Davis Veterinary Genetics Laboratory.

### Incidence of melanoma in slow greying horses

All 16 Connemara horses included in the study (10 slow grey (*G1/G2*) and 6 fast grey (*G1/G3*) offspring descending from the stallion Hagens D'Arcy with non-grey mothers) were examined clinically by a veterinarian regarding the presence of melanoma. Two of the fast greying offspring (11 and 13 years old, respectively) were found to be affected by melanoma on the underside of the tail, while the rest of the offspring were negative. As the sampled horses were relatively young (half were between 0 and 7 years old) and the total number low, it was not possible to draw any conclusions regarding the melanoma incidence for fast greys and slow greys respectively in the Connemara pony based on this material.

To estimate the incidence of melanoma in slow greying Connemara ponies, we did a study of 25 slow greying horses between 15 and 35 years old (Supplementary Table 4). Four of the horses were included in the mapping study as offspring of Hagens D'Arcy and were examined by a veterinarian for melanoma at the time of blood sampling. The examination of the other 21 horses was carried out as an interview where a veterinarian asked neutral questions to owners or breeders of the horses if they had observed visible melanomas in the horses from the age of 15 onwards. Those who answered the questions were experienced breeders or horse keepers who were judged to have good knowledge of the symptoms of melanoma in horses. None of the respondents stated that they had observed melanoma in the horses in question.

**Reporting summary**

Further information on research design is available in the Nature Portfolio Reporting Summary linked to this article.

## Data availability

The sequence data generated in this study have been deposited in the NCBI database (https://www.ncbi.nlm.nih.gov/bioproject/PRJNA1035120/). The GWAS summary statistics are available at: https://github.com/LeifAnderssonLab/Equine_late_greying. All other data supporting the findings described in this manuscript are available in the article and its Supplementary Information files.

## Code availability

The analyses of data have been carried out with publicly available software and all are cited in the Methods section. Code associated with bioinformatic analyses is available at: https://github.com/LeifAnderssonLab/Equine_late_greying.

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

## Acknowledgements

We thank all horse owners, in particular the owners of Hagens D'Arcy and his offspring that have allowed us to sample their horses and consented to research. We also thank Dr. Robert Grahn, Thea Ward, Shayne Hughes, Elizabeth Esdaile, and Tytti Vanhala for their technical assistance and Sara Yousefi Taemeh for help with illustrations. The project was financially supported by Vetenskapsrådet (2017-02907; to LA), Knut and Alice Wallenberg Foundation (KAW 2023.0160; to LA) and the Austrian Science Fund (FWF P35840; to LA). Short-read Illumina sequencing was performed by the SNP&SEQ Technology Platform in Uppsala. The facility is part of the National Genomics Infrastructure (NGI) Sweden and Science for Life Laboratory. The SNP&SEQ Platform is also supported by the Swedish Research Council and the Knut and Alice Wallenberg Foundation. The computational infrastructure was provided by the National Academic Infrastructure for Supercomputing in Sweden, partially funded by the Swedish Research Council through grant agreement no. 2022-06725.

## Author contributions

LA and GL conceived the study. C-JR was responsible for nCATS, nanopore adaptive sampling and bioinformatic analyses. MH, BWD and C-JR designed and MH, BWD evaluated the nCATS experiments. RN performed ddPCR analysis and prepared sequencing libraries for the Connemara pony family. RF performed ddPCR analysis for the Japanese Thorougbred family. MB collected family material and phenotype data from Connemara ponies. RB, AK, and SJU were responsible for *Grey* genotyping across breeds. HO, KS, AO, and TT were responsible for collection of the Japanese Thoroughbred family. LA wrote the manuscript with input from other authors. All authors approved the manuscript before submission.

## Funding

## Competing interests

R.R. Bellone, A. Kallenberg, and S. J'Usrey are affiliated with the UC Davis Veterinary Genetics Laboratory, which provides genetic diagnostic tests in horses and other species. The other authors declare no competing interest.
