## [Peer Review File · Nature Communications]

An intronic copy number variation in *Syntaxin 17* determines speed of greying and melanoma incidence in Grey horsesREVIEWER COMMENTS

Reviewer #1 (Remarks to the Author):

The manuscript by Rubin et al. employs a forward genetic approach in horse pedigrees to refine the molecular basis for progressive greying coat color, which is often coincident with melanomas. Prior work by this group reported the causal variant as a 4.6 kb duplication that induces melanocyte-specific expression of two nearby genes. The current study reveals that copy number variation affects the rate of greying and the incidence of melanoma. Alleles with two (G2) or three (G3) tandem copies of the 4.6 kb segment cause slow greying with no melanoma, or fast greying with high melanoma incidence, respectively.

The manuscript is concise, clearly written, and engaging. It convincingly identifies copy number variation that contributes to the modified expression of a captivating equine trait, with relevance to human health and disease. For this reason, the study will be of broad appeal.

There were several aspects of manuscript that need clarification or correction.

(1) The authors should explicitly state which individuals are used in the GWAS and add a supplemental table, split by phenotypic class, with the following information for each individual: pedigree/generation (Connemara or Japanese thoroughbred), inferred copy number variant (CNV) genotype, and copy number estimates from WGS and/or ddPCR.

The GWAS cohorts are reported inconsistently in the manuscript:

Line 126 (Results): “we carried out whole genome sequencing of a total of 13 slow greying and 7 fast greying horses.”

whereas...

Line 354 (Methods): “GWAS was performed...with phenotypes of 13 fast greying individuals...and phenotypes of 8 slow greying individuals.”

Not only are the cohort sizes inverted, but the total number of individuals are inconsistent - 20 in the results section and 21 in the methods section. The animals presumably used for GWAS include Hagens D’Arcy (the sire), his 16 offspring, and 5 individuals from a Japanese thoroughbred pedigree - 22 individuals, not 20 or 21. A cohort size of 22 is also consistent with the number of sequenced individuals, displayed as rows in Figure 3D.

(2) Additions or changes to genotyping and the GWAS:

The authors should report the total number of discovered variants and the number of variants sites

used for the GWAS. The number of GWAS variants is important for establishing a genome-wide significance threshold, which should be indicated in Figure 3A.

Figure 3B does not provide sufficient resolution to visualize patterns of association near the suspected causal variant. A higher resolution (~1-2Mb) view should be presented. The authors report a 350 kb IBD interval shared by the G2 and G3 alleles, which should result in a trough in the GWAS peak surrounding the previously reported Grey locus. The GWAS signal therefore should be driven by flanking variants (outside the IBD interval) in LD with the CNV, unless there are allele specific variants in the IBD interval. The position of the IBD interval should be indicated in the revised plot.

The perfect association between copy number and rate of greying provides strong evidence of causality, but the authors should report any relevant genomic characteristics (coding/noncoding, ancestral/derivative, evolutionary and predicted functional constraint) useful to exclude a causal role for other polymorphisms exhibiting perfect association.

Figure 3C: This panel shows only the genotyping data for the Connemara horses. Was ddPCR genotyping done for the 5 Japanese thoroughbred horses? The authors mention in the text that the G2 and G3 alleles in the Japanese pedigree are perfectly associated with the rate of greying, but this data is not presented in a figure or table. While the correlation between genotyping methods (Figure 3C) is satisfying and important, it is more important to show, in a main figure or table, that the CNV genotypes for all GWAS individuals are perfectly associated with the rate of greying.

Figure 3D: The figure legend states that nucleotide diversity is measured in 100 kb windows, but the window size in the panel is 50 kb, not 100 kb.

The authors indicate that the $-\log_{10}(P)$ values were changed in figures 3A and 3B to better scale the plots. The unmodified P-values should be reported because they are a statistical measure of the strength of association. It is hard to believe, given the small sample size, that P-values for these variants “were more than a hundred orders of magnitude lower than those of associated SNPs with a non-perfect association”. The authors should also report the exact P-value for association of the CNV.

(3) The genotype data should be used to address whether Connemara and Japanese thoroughbred horses share the 350 kb IBD interval by inferring haplotypes across the region. Within each pedigree, are there IBD interval polymorphisms private to G2 or G3 alleles? If so, are the G2 or G3 specific polymorphisms shared across breeds or shared with the G1 haplotype? This line of investigation could offer insight into evolutionary relationship of alleles within and across breeds.

The presumed low recombination rates and potential recombination-mediated variation in copy number is interesting. Given that stallions often sire many offspring, do studbook or pedigree records exist that might pinpoint exceptional non-greying offspring from G3/G3 sires? Similarly, have the authors considered applying their single-molecule sequencing assay to measure rates of unequal crossover events (and recombination) in sperm? The same assay could also be used to

measure copy number heterogeneity in melanomas. Both applications are potentially interesting lines of investigation for future studies.

(4) Is phenotype data available for horses from other breeds that have 3 (2 + 1) copies of the CNV? If so, the authors should report phenotype information from these horses to more thoroughly address whether the slower rate of greying is observed in different breeds and genetic backgrounds.

(5) The methylation analysis should be described in more detail in Methods? How many CpG sites are recorded, and does the data plotted in Figure 4C represent a windowed average or kernel density estimate of all sites? Is there a precedent for detection of cell-type specific enhancer methylation signals in unrelated cell lineages (i.e. melanocyte enhancers in blood cells)?

(6) Figure 1: What is the purpose of the two images above the timeline? Also, the line connecting the first image of the slow greying horse to the timeline is missing.

Reviewer #2 (Remarks to the Author):

The greying with age coat colour, designated as the Grey (G) locus, in the horse has been a challenge to understand with a diverse phenotypic presentation and genetic mechanism(s) underlying it. The causal mutation for Grey was originally identified as a 4.6 kb duplication in intron 6 of the Syntaxin 17 (STX17) gene. This large duplication has meant that proper characterisation and genotyping of this locus has been problematic, and it has only recently been determined using more advanced methods (droplet digital PCR and now sequence coverage/nanopore sequencing) that the most frequent variant allele for Grey is actually a triplication of the 4.6kb region. Thus, there are three alleles for the Grey locus in the horse: G1 as WT, G2 with two copies and G3 with three copies of the repeat.

The unusual presentation of the greying phenotype is that the speed of coat greying varies considerably among Grey horses, and it was thought that the Grey genotype plays a role. Indeed, Grey homozygotes grey faster than Grey heterozygotes. There is also variability in the development of melanocytic lesions that also correspond with the speed of greying. The aim of the present study was to explore the genetic basis for this phenotypic variability.

The phenotype seen in Connemara ponies with two distinct types of greying are noted, fast greying and slow greying, and the starting point for the study. Segregation of the fast and slow presentations in the pedigree of the fast greying sire Hagens D'Arcy showed a Mendelian inheritance pattern with 4.6 kb copy number. This was also confirmed in a small family of Japanese thoroughbred horses. Based on these results the authors propose a new nomenclature for Grey as G1 - one copy, WT; G2 – two copies, slow greying; G3 – three copies, fast greying. Pedigree analysis identified one horse as a putative G2/G2 homozygote, with a phenotype intermediate between fast and slow presentations

(at age 11 years), without melanoma. In genotyping across breeds, the most common genotype was for four copies of the 4.6kb repeat, most likely representing the G1/G3 heterozygote genotype.

The data is quite convincing and allow the authors to conclude that a remarkable dosage effect whereby the G3 allele is associated with fast greying and high incidence of melanoma, whereas G2 is associated with slow greying and low incidence of melanoma. I have one reservation with their conclusion in the Discussion (lines 244 to 246),

“... G1/G3 shows fast greying and high incidence of melanoma, and G3/G3 homozygotes grey very fast and show a very high incidence of melanoma.”

There is no quantitative evidence presented in this paper to support this conjecture about the G3/G3 homozygote vs the G1/G3 heterozygote phenotype. It may be true, but it has not been demonstrated/analysed in their data?

Minor comments:

1. In the sentence beginning on line 237, “Although this study explains a considerable portion of the heterogeneity in speed of greying, it is likely that additional genetic variation elsewhere in the genome contributes to variation in the speed of greying and these additional loci may also contribute to melanoma risk”

The authors may wish to consider the citation of,
Druml T, Brem G, Horna M, Ricard A, Grilz-Seger G
DPF3, A Putative Candidate Gene For Melanoma Etiopathogenesis in Gray Horses.
J Equine Vet Sci. 2022 Jan;108:103797

2. In Figure 4c, the authors could consider including the plot of the results for the one copy of the 4.6kb allele that is schematically illustrated at the bottom of the panel.

3. Figure 4c legend, “DNA methylation pattern associated with the G2 (red) and G3 (blue) alleles.”
The plot for G2 is black and G3 is grey in the copies of the panel I see.

4. In Table 1, the authors could consider putting in the allele frequency for G1, as they have for G2 and G3

Reviewer #3 (Remarks to the Author):

This manuscript entitled “An intronic copy-number variation in Syntaxin 17 determines speed of greying and melanoma incidence in Grey horses” by Andersson, L. and colleagues [Nature Communications MS# NCOMMS-23-54485-T] describes a wonderful global genetic study on the greying with age phenotype in horses. This phenotype involves loss of hair pigmentation whereas

skin pigmentation is not reduced. Remarkably, such horses also show a predisposition to melanoma. Whilst it was established previously that the trait was initially associated with a duplication CNV of a 4.6 kb intronic sequence in Syntaxin 17, the speed of greying varies considerably amongst Grey horses. In this manuscript, the authors demonstrate the difference between two different Grey alleles, G2 carrying two tandem copies of the duplicated sequence and G3 carrying three copies of that segment that is duplicated. The latter is by far the most common allele most likely due to strong selection for the striking white phenotype. Interestingly, in this study, the results reveal a remarkable dosage effect with the G3 allele associated with fast greying and high incidence of melanoma and G2 is associated with slow greying and low incidence of melanoma. Furthermore, epigenetic analysis of long-read sequencing data documents a drastic reduction in DNA methylation of the region and the sequence harbors MITF binding sites. This copy number expansion transforms a weak enhancer to a strong melanocyte-specific enhancer that underlies hair greying (G2 and G3) and dramatically elevated risk of melanoma (G3 only). This is a fascinating study and likely to be of interest to a broad spectrum of readers of this journal. Lots of lessons to be learned here!

There are however a few things that the co-authors might want to consider:

Intronic CNV, greying horses, dosage, melanoma risk

- i) Please assign omim.org MIM# when describing human disease trait or disease susceptibility: e.g. (WS2A; MIM:193510); Tiet albinism-deafness syndrome (TADS; MIM:103500), melanoma, cutaneous malignant susceptibility (CMM8; MIM: 6144560). This will be helpful to all your readers to ensure talking about same rare disease trait locus in the human organism.
 - ii) Might they comment on any overlap of human TADS (MIM: 103500) with the mouse knockout model?
 - iii) Regarding the hypothesis ‘...that the initial mutation can be traced to a single mutational event that happened prior to horse domestication’ - what are the breakpoint junctions and do the authors care to ‘speculate’ on the mutational mechanism in the clan?
 - iv) I do think/suggest it is important that the long read data allows ‘phasing’ of sequence variation and may help the reader if that is more explicitly stated.
 - v) Suggest perhaps ddPCR of breakpoint junction in G3 vs G2 may be informative for the tandem rearrangement head-tail model?
 - vi) Does the G2/G2 homozygote make this ‘intermediate slow/fast greying’ a “codominant” or “semidominant” trait? Just curious of how the allele is considered?
 - vii) G2/G2 had “...no visible melanoma”. At what age was this ‘assayed’
 - viii) Regarding Discussion and possibility of NAHR driven ‘expansion/contraction’ for derivation of G3 vs G2, whilst this is a reasonable hypothesis knowing the breakpoint junctions and whether microhomology involved or repetitive sequences, etc. might be helpful as noted in Liu, et al 2014 Am J Hum Gen 94: 462-469 and BioArchives Grochowski, et al.
- Jim Lupski

Response to reviewers

We thank the reviewers for their constructive criticism of our paper. These are the most important changes in the revised version:

1: We have added ddPCR data for the Japanese Thoroughbred pedigree. This led to the discovery that the G2 allele arose by copy number contraction in a G3/G3 parent. This result is presented in the new Fig 5 and in the new section: “The birth of a G2 allele” (Line 214-230).

2. We have added Nanopore sequencing of a G2 homozygote which allowed us to deduce the G2 haplotype and we show that the G2 haplotype in Swedish Connemara and Japanese Thoroughbred horses show no IBD region outside the 350 kb being IBD among all G2 and G3 haplotypes. This result is consistent with our observation that the G2 allele arose by mutation very recently in the Japanese horses.

3. We have performed a more detailed analysis of the sequence and in fact identified microhomology between the flanking sequences which may have contributed to the initial Grey mutation.

Our responses to the reviewers’ comments are given below in bold.

Reviewer #1 (Remarks to the Author):

The manuscript by Rubin et al. employs a forward genetic approach in horse pedigrees to refine the molecular basis for progressive greying coat color, which is often coincident with melanomas. Prior work by this group reported the causal variant as a 4.6 kb duplication that induces melanocyte-specific expression of two nearby genes. The current study reveals that copy number variation affects the rate of greying and the incidence of melanoma. Alleles with two (G2) or three (G3) tandem copies of the 4.6 kb segment cause slow greying with no melanoma, or fast greying with high melanoma incidence, respectively.

The manuscript is concise, clearly written, and engaging. It convincingly identifies copy number variation that contributes to the modified expression of a captivating equine trait, with relevance to human health and disease. For this reason, the study will be of broad appeal.

There were several aspects of manuscript that need clarification or correction.

(1) The authors should explicitly state which individuals are used in the GWAS and add a supplemental table, split by phenotypic class, with the following information for each individual: pedigree/generation (Connemara or Japanese thoroughbred), inferred copy number variant (CNV) genotype, and copy number estimates from WGS and/or ddPCR.

>>>These data are now presented in Supplementary Table 1

The GWAS cohorts are reported inconsistently in the manuscript:

Line 126 (Results): “we carried out whole genome sequencing of a total of 13 slow greying and 7 fast greying horses.”

>>>This mistake has been corrected. This should be 9 fast greying horses and 13 slow greying horses. We have now changed this in the manuscript text.

whereas...

Line 354 (Methods): “GWAS was performed...with phenotypes of 13 fast greying individuals...and phenotypes of 8 slow greying individuals.”

>>>This should be 8 fast greying horses (because the pedigree sire was not included in GWAS) and 13 slow greying horses. We have now changed this in the manuscript text.

Not only are the cohort sizes inverted, but the total number of individuals are inconsistent - 20 in the results section and 21 in the methods section. The animals presumably used for GWAS include Hagens D’Arcy (the sire), his 16 offspring, and 5 individuals from a Japanese thoroughbred pedigree - 22 individuals, not 20 or 21. A cohort size of 22 is also consistent with the number of sequenced individuals, displayed as rows in Figure 3D.

>>>The reviewer is right. These were 22 individuals including the Connemara sire.

(2) Additions or changes to genotyping and the GWAS:

The authors should report the total number of discovered variants and the number of variants sites used for the GWAS. The number of GWAS variants is important for establishing a genome-wide significance threshold, which should be indicated in Figure 3A.

>>>We obtained 8,426,579 SNPs after filtering, removing all positions with missing genotypes and SNPs with minor allele frequencies <0.1, and used these in GWAS. This means that none of the SNPs reached genome-wide significance, top SNPs showing complete association get a *P*-value (Fisher’s exact test) of 4.9×10^{-6} . However, we conclude on Line 140-141 that “the fact that 394 out of the 397 most strongly associated SNPs map within 10 Mb of *STX17* provides compelling support for this location”.

Figure 3B does not provide sufficient resolution to visualize patterns of association near the suspected causal variant. A higher resolution (~1-2Mb) view should be presented. The authors report a 350 kb IBD interval shared by the G2 and G3 alleles, which should result in a trough in the GWAS peak surrounding the previously reported Grey locus. The GWAS signal therefore should be driven by flanking variants (outside the IBD interval) in LD with the CNV, unless there are allele specific variants in the IBD interval. The position of the IBD interval should be indicated in the revised plot.

>>>This is now shown in Supplementary Figure 4a where a zoom-in on the region chr25:5-8Mb is presented, and the reviewer is right there is a paucity of informative sites in the IBD region shared by all G2 and G3 haplotypes.

The perfect association between copy number and rate of greying provides strong evidence of causality, but the authors should report any relevant genomic characteristics (coding/noncoding, ancestral/derivative, evolutionary and predicted functional constraint) useful to exclude a causal role for other polymorphisms exhibiting perfect association.

>>>In addition to the CNV, six SNPs show perfect association. Their locations, perceived consequence to gene models and levels of evolutionary constraint are shown in Supplementary Table 2. None of these occur in coding sequences or at highly conserved sites.

Figure 3C: This panel shows only the genotyping data for the Connemara horses. Was ddPCR genotyping done for the 5 Japanese thoroughbred horses? The authors mention in the text that the G2 and G3 alleles in the Japanese pedigree are perfectly associated with the rate of greying, but this data is not presented in a figure or table. While the correlation between genotyping methods (Figure 3C) is satisfying and important, it is more important to show, in a main figure or table, that the CNV genotypes for all GWAS individuals are perfectly associated with the rate of greying.

>>>Figure 3C has been updated and now includes the Japanese horses. The ddPCR and WGS data for each Connemara and Thoroughbred individual are also shown in the new Supplementary Table 1. The addition of ddPCR data for the Japanese pedigree led to the discovery of how the G2 allele in this family originated by copy number contraction from a G3 allele. The data is presented in Figure 5 and a new section “The birth of a G2 allele” has been added to the Results (Line 214-230). The Discussion has also been modified to acknowledge the implications of this important finding.

Figure 3D: The figure legend states that nucleotide diversity is measured in 100 kb windows, but the window size in the panel is 50 kb, not 100 kb.

>>>These were overlapping sliding windows of 100kb, moving 50kb forward. This figure has become Supplementary Fig 4b. In the revised version of Figure 3d we explore sequence identities among all the G2 and G3 haplotypes sequenced in this study. This new analysis reveals that the G2 and G3 haplotypes share an IBD region between chr25:6.45-6.8 Mb.

The authors indicate that the $-\log_{10}(P)$ values were changed in figures 3A and 3B to better scale the plots. The unmodified P-values should be reported because they are a statistical measure of the strength of association. It is hard to believe, given the small sample size, that P-values for these variants “were more than a hundred orders of magnitude lower than those of associated SNPs with a non-perfect association”. The authors should also report the exact P-value for association of the CNV.

>>>We had previously used the software vcf2gwas for GWAS analysis. We now decided to perform GWAS using PLINK, applying the Fisher’s exact test assuming a dominant mode of inheritance. This new analysis identified 6 SNP positions with perfect association to speed of greying, and the P-values of these were 4.9×10^{-6} , which was also the P-value obtained for association between CNV copy numbers and speed of greying.

(3) The genotype data should be used to address whether Connemara and Japanese thoroughbred horses share the 350 kb IBD interval by inferring haplotypes across the region. Within each pedigree, are there IBD interval polymorphisms private to G2 or G3 alleles? If so, are the G2 or G3 specific polymorphisms shared across breeds or shared with the G1 haplotype? This line of investigation could offer insight into evolutionary relationship of alleles within and across breeds.

>>>We performed a new analysis, leveraging Oxford nanopore sequencing data of a G2/G2 individual. For each G1/G2 and G1/G3 individual where NGS data was available we calculated the fraction of opposite homozygote calls observed in relation to the genotype of the G2/G2 individual. This data is presented in the new Figure 3d and reveals that all individuals lack opposite homozygote SNP calls for the 350 kb region, consistent with this region being IBD between G2 and G3 haplotypes. We also show that all G2 haplotypes from the Connemara breeds shares a > 5 Mb-long region of IBD suggesting that they all trace back to a relatively recent common ancestor. In contrast, the sequence identity between G2 haplotypes from Connemara and Japanese Thoroughbred horses break down outside the 350 kb IBD region shared by all G2 and G3 haplotypes, suggesting that the G2 haplotypes from the two breeds do not have a recent common ancestry. This is of course confirmed by the fact that the G2 in the Japanese pedigree arose from a G3 allele.

The presumed low recombination rates and potential recombination-mediated variation in copy number is interesting. Given that stallions often sire many offspring, do studbook or pedigree records exist that might pinpoint exceptional non-greying offspring from G3/G3 sires? Similarly, have the authors considered applying their single-molecule sequencing assay to measure rates of unequal crossover events (and recombination) in sperm? The same assay could also be used to measure copy number heterogeneity in melanomas. Both applications are potentially interesting lines of investigation for future studies.

>>> As mentioned above, the addition of ddPCR data for the Japanese Thoroughbred pedigree resulted in the discovery of the birth of G2 allele by copy number contraction during meiosis of the G3/G3 dam! So, we know now that it occurs. Hopefully this paper will make horse breeders more observant to the possible occurrence of such mutations so that we can get information on how common it is.

It is certainly an interesting possibility to try to use our method to estimate the mutation frequency by analyzing sperm. Nanopore sequencing on a Oxford nanopore PromethION flow cell would yield 15-30x coverage of a sperm sample. Using either of the two long-read enrichment methods, i.e. Cas9 capture or adaptive sampling would be expected to yield approximately a 5-10 fold enrichment, i.e 75-150x coverage. Useful reads would need to be approximately >20kb in length to span the entire CNV and to be well-anchored in unique sequence on each side. We estimate that 50-70% of reads would be above 20 kb in size, rendering, a best-case scenario of 150x coverage x 0.7 = 100 times coverage of useful reads. Thus, if the incidence of unequal crossover events is prevalent (several percent), a single PromethION run using targeted enrichment may be sufficient. If the incidence is low, for example 0.1%, then 10 PromethION runs would be needed just to observe one unequal crossover event. Therefore, the suggested experiment, although very interesting, would be very expensive at present. We prefer to await methodological advances before investigating this.

(4) Is phenotype data available for horses from other breeds that have 3 (2 + 1) copies of the CNV? If so, the authors should report phenotype information from these horses to more

thoroughly address whether the slower rate of greying is observed in different breeds and genetic backgrounds.

>>>No, unfortunately not, but in the revised version we have added data for two fast Grey G1/G3 Quarter horses in Supplementary Fig. 7.

(5) The methylation analysis should be described in more detail in Methods? How many CpG sites are recorded, and does the data plotted in Figure 4C represent a windowed average or kernel density estimate of all sites? Is there a precedent for detection of cell-type specific enhancer methylation signals in unrelated cell lineages (i.e. melanocyte enhancers in blood cells)?

>>> Methylation status was assessed at 41 CpG sites for each copy of the duplication and at 15 and 17 CpG sites flanking the duplicated fragment up-and downstream, respectively. In Fig 4C, the per-site 5mC methylation fractions were averaged in sliding windows of 5 adjacent CpGs, moving forward one CpG site at a time. This is now explained on Line 476-482.

As regards cell specificity of the DNA methylation pattern for this enhance, we have noted that an extensive analysis of human ENCODE data reported that intronic “tissue-specific” enhancers that regulates non-host genes tend to be less tissue-specific than intronic “tissue-specific” enhancers that only regulates the host gene (Ref: *Genome Res* 2021, 31:1325-1335 (doi: [10.1101/gr.270371.120](https://doi.org/10.1101/gr.270371.120)). This appears consistent with our result because we have previously reported that the Grey mutation leads to upregulated expression of both *STX17* and *NR4A3* (Ref 1 in our paper).

(6) Figure 1: What is the purpose of the two images above the timeline? Also, the line connecting the first image of the slow greying horse to the timeline is missing.

>>>Figure 1 has been corrected and now all images of horses are connected to the correct position on the timeline indicating their age when the picture was taken.

Reviewer #2 (Remarks to the Author):

The greying with age coat colour, designated as the Grey (G) locus, in the horse has been a challenge to understand with a diverse phenotypic presentation and genetic mechanism(s) underlying it. The causal mutation for Grey was originally identified as a 4.6 kb duplication in intron 6 of the Syntaxin 17 (*STX17*) gene. This large duplication has meant that proper characterisation and genotyping of this locus has been problematic, and it has only recently been determined using more advanced methods (droplet digital PCR and now sequence coverage/nanopore sequencing) that the most frequent variant allele for Grey is actually a triplication of the 4.6kb region. Thus, there are three alleles for the Grey locus in the horse: G1 as WT, G2 with two copies and G3 with three copies of the repeat.

The unusual presentation of the greying phenotype is that the speed of coat greying varies considerably among Grey horses, and it was thought that the Grey genotype plays a role. Indeed, Grey homozygotes grey faster than Grey heterozygotes. There is also variability in the development of melanocytic lesions that also correspond with the speed of greying. The

aim of the present study was to explore the genetic basis for this phenotypic variability.

The phenotype seen in Connemara ponies with two distinct types of greying are noted, fast greying and slow greying, and the starting point for the study. Segregation of the fast and slow presentations in the pedigree of the fast greying sire Hagens D'Arcy showed a Mendelian inheritance pattern with 4.6 kb copy number. This was also confirmed in a small family of Japanese thoroughbred horses. Based on these results the authors propose a new nomenclature for Grey as G1 – one copy, WT; G2 – two copies, slow greying; G3 – three copies, fast greying. Pedigree analysis identified one horse as a putative G2/G2 homozygote, with a phenotype intermediate between fast and slow presentations (at age 11 years), without melanoma. In genotyping across breeds, the most common genotype was for four copies of the 4.6kb repeat, most likely representing the G1/G3 heterozygote genotype.

The data is quite convincing and allow the authors to conclude that a remarkable dosage effect whereby the G3 allele is associated with fast greying and high incidence of melanoma, whereas G2 is associated with slow greying and low incidence of melanoma. I have one reservation with their conclusion in the Discussion (lines 244 to 246),

“... G1/G3 shows fast greying and high incidence of melanoma, and G3/G3 homozygotes grey very fast and show a very high incidence of melanoma.”

There is no quantitative evidence presented in this paper to support this conjecture about the G3/G3 homozygote vs the G1/G3 heterozygote phenotype. It may be true, but it has not been demonstrated/analysed in their data?

>>>This was demonstrated in our previous paper Rosengren Pielberg et al. (2008). We have modified the text to make it is clear that this statement is based on our previous study of 800 Lippizaner horses.

Minor comments:

1. In the sentence beginning on line 237, “Although this study explains a considerable portion of the heterogeneity in speed of greying, it is likely that additional genetic variation elsewhere in the genome contributes to variation in the speed of greying and these additional loci may also contribute to melanoma risk”

The authors may wish to consider the citation of,

Druml T, Brem G, Horna M, Ricard A, Grilz-Seger G

DPF3, A Putative Candidate Gene For Melanoma Etiopathogenesis in Gray Horses.

J Equine Vet Sci. 2022 Jan;108:103797

>>>We have added this reference.

2. In Figure 4c, the authors could consider including the plot of the results for the one copy of the 4.6kb allele that is schematically illustrated at the bottom of the panel.

>>>In addition to the G2/G3 Connemara sire shown in Fig 4C we now also include methylation data derived from nanopore sequencing for a G2/G2 individual and for two G1/G3 individuals (Supplementary Figure 7)

3. Figure 4c legend, “DNA methylation pattern associated with the G2 (red) and G3 (blue) alleles.” The plot for G2 is black and G3 is grey in the copies of the panel I see.

>>>This mistake has now been corrected.

4. In Table 1, the authors could consider putting in the allele frequency for G1, as they have for G2 and G3

>>>We don't think this is needed because G1 can easily be calculated as 1 – G2 – G3. More importantly, the purpose of this table is to estimate the relative frequency of the G2 and G3 haplotypes but we are sure that these are not unbiased estimates of the allele frequencies because the data are based on horse owners submitting samples for genotyping and it is very likely that in most cases, they expect that the horse may have a Grey allele. Thus, the allele frequencies for G2 and G3 are expected to be inflated.

Reviewer #3 (Remarks to the Author):

This manuscript entitled “An intronic copy-number variation in Syntaxin 17 determines speed of greying and melanoma incidence in Grey horses” by Andersson, L. and colleagues [Nature Communications MS# NCOMMS-23-54485-T] describes a wonderful global genetic study on the greying with age phenotype in horses. This phenotype involves loss of hair pigmentation whereas skin pigmentation is not reduced. Remarkably, such horses also show a predisposition to melanoma. Whilst it was established previously that the trait was initially associated with a duplication CNV of a 4.6 kb intronic sequence in Syntaxin 17, the speed of greying varies considerably amongst Grey horses. In this manuscript, the authors demonstrate the difference between two different Grey alleles, G2 carrying two tandem copies of the duplicated sequence and G3 carrying three copies of that segment that is duplicated. The latter is by far the most common allele most likely due to strong selection for the striking white phenotype. Interestingly, in this study, the results reveal a remarkable dosage effect with the G3 allele associated with fast greying and high incidence of melanoma and G2 is associated with slow greying and low incidence of melanoma. Furthermore, epigenetic analysis of long-read sequencing data documents a drastic reduction in DNA methylation of the region and the sequence harbors MITF binding sites. This copy number expansion transforms a weak enhancer to a strong melanocyte-specific enhancer that underlies hair greying (G2 and G3) and dramatically elevated risk of melanoma (G3 only). This is a fascinating study and likely to be of interest to a broad spectrum of readers of this journal. Lots of lessons to be learned here!

There are however a few things that the co-authors might want to consider:

Intronic CNV, greying horses, dosage, melanoma risk

i) Please assign omim.org MIM# when describing human disease trait or disease susceptibility: e.g. (WS2A; MIM:193510); Tiet albinism-deafness syndrome (TADS; MIM:103500), melanoma, cutaneous malignant susceptibility (CMM8; MIM: 6144560). This will be helpful to all your readers to ensure talking about same rare disease trait locus in the human organism.

>>>We have added WS2A; MIM:193510 and TADS; MIM:103500 to this sentence.

ii) Might they comment on any overlap of human TADS (MIM: 103500) with the mouse knockout model?

>>>We think that is out of the scope of the present study, since Greying with age in horses is not caused by mutations in MITF.

iii) Regarding the hypothesis ‘...that the initial mutation can be traced to a single mutational event that happened prior to horse domestication’ - what are the breakpoint junctions and do the authors care to ‘speculate’ on the mutational mechanism in the clan?

>>>We thank the reviewer for this constructive comment. We now report the breakpoint junctions in Supplementary Fig. 6 and mention in the main text (Line 197-201) that there is a 6 bp micro-homology immediately flanking the breakpoints on each side which may have contributed to the initial duplication of the CNV sequence.

iv) I do think/suggest it is important that the long read data allows ‘phasing’ of sequence variation and may help the reader if that is more explicitly stated.

>>>We have added this comment on Line 185-186.

v) Suggest perhaps ddPCR of breakpoint junction in G3 vs G2 may be informative for the tandem rearrangement head-tail model?

>>>We have previously described a normal PCR test amplifying the duplication breakpoint in fast greying horses (see reference 1: Rosengren Pielberg et al., 2008). The Nanopore sequence data presented here confirm the head-tail model for both the G2 and G3 haplotypes. Furthermore, such a ddPCR assay would still not distinguish the G1/G3 and G2/G2 genotypes.

vi) Does the G2/G2 homozygote make this ‘intermediate slow/fast greying’ a “codominant” or “semidominant” trait? Just curious of how the allele is considered?

>>>A semi-dominant inheritance of hair greying and incidence of melanoma is well established for the G3 allele based on our previous publication as explained on Line 297-299. However, since we only have data on a single G2/G2 individual it is too early to conclude whether also the G2 allele show semi-dominance although this appears likely.

vii) G2/G2 had “...no visible melanoma”. At what age was this ‘assayed’

>>>We state on Line 262-264 and on Line 302-304 that this horse had no visible melanoma at 11 years of age.

viii) Regarding Discussion and possibility of NAHR driven ‘expansion/contraction’ for derivation of G3 vs G2, whilst this is a reasonable hypothesis knowing the breakpoint junctions and whether microhomology involved or repetitive sequences, etc. might be helpful as noted in Liu, et al 2014 Am J Hum Gen 94: 462-469 and BioArchives Grochowski, et al.

>>>As indicated in our comments to Reviewer 1, we have now discovered an example of a copy number contraction resulting in the birth of a G2 allele from a G3 allele. Unfortunately, we could not determine whether this occurred by an intra- or interchromosomal event because the parent showed complete homozygosity over a 14 Mb region, see the new section “The birth of a G2 allele” (Line 214-230).

Jim Lupski
>>>**Thanks!**

REVIEWERS' COMMENTS

Reviewer #1 (Remarks to the Author):

The authors have satisfactorily addressed my comments in the revised manuscript. The identification and characterization of a revertant allele from the Japanese thoroughbred pedigree adds compelling support for the specific effect of copy number variation on speed of greying. I endorse the manuscript for publication.

Reviewer #3 (Remarks to the Author):

Fantastic science - interesting biology and genetics.